# A Universal Landslide Detection Method in Optical Remote Sensing Images Based on Improved YOLOX

**Heyi Hou** [1], **Mingxia Chen** [1,*], **Yongbo Tie** [2] **and Weile Li** [3]

1 College of Mechanical and Control Engineering, Guilin University of Technology, Guilin 541006, China
2 Chengdu Center of China Geological Survey, Chengdu 610081, China
3 State Key Laboratory of Geohazard Prevention and Geoenvironment Protection, Chengdu University of Technology, Chengdu 610059, China
* Correspondence: wjunt@sohu.com

**Abstract:** Using deep learning-based object detection algorithms for landslide hazards detection is very popular and effective. However, most existing algorithms are designed for landslides in a specific geographical range. This paper constructs a set of landslide detection models YOLOX-Pro, based on the improved YOLOX (You Only Look Once) target detection model to address the poor detection of complex mixed landslides. Wherein the VariFocal is used to replace the binary cross entropy in the original classification loss function to solve the uneven distribution of landslide samples and improve the detection recall; the coordinate attention (CA) mechanism is added to enhance the detection accuracy. Firstly, 1200 historical landslide optical remote sensing images in thirty-eight areas of China were extracted from Google Earth to create a mixed sample set for landslide detection. Next, the three attention mechanisms were compared to form the YOLOX-Pro model. Then, we tested the performance of YOLOX-Pro by comparing it with four models: YOLOX, YOLOv5, Faster R-CNN, and Single Shot MultiBox Detector (SSD). The results show that the YOLOX-Pro(m) has significantly improved the detection accuracy of complex and small landslides than the other models, with an average precision (AP0.75) of 51.5%, APsmall of 36.50%, and ARsmall of 49.50%. In addition, optical remote sensing images of a 12.32 km$^2$ group-occurring landslides area located in Mibei village, Longchuan County, Guangdong, China, and 750 Unmanned Aerial Vehicle (UAV) images collected from the Internet were also used for landslide detection. The research results proved that the proposed method has strong generalization and good detection performance for many types of landslides, which provide a technical reference for the broad application of landslide detection using UAV.

**Keywords:** landslide detection; optical remote sensing image; deep learning; attention mechanism; unmanned aerial vehicle; YOLOX

## 1. Introduction

Landslide is one of the common geological disasters in mountainous areas, which can seriously endanger human life, and property and damage the natural environment. Due to global climate change, the frequency and intensity of landslide disasters have increased in recent years. For example, the 2018 Baige landslide on the Jinsha River in Tibet blocked the Jinsha River from forming a dammed lake, which caused disasters hundreds of kilometers downstream after the dammed lake burst [1]. For such landslide disasters occurring in uninhabited regions, using satellite images to quickly locate and extract landslide feature information and delineate its potential impact area is significant for timely disaster diagnosis, post-disaster rescue, and landslide database establishment.

Traditional landslide hazard analysis includes field exploration and manual visual interpretation methods using remote sensing data [2,3]. Huang et al. [4] identified 11,308 geological hazards triggered by the 2008 Wenchuan earthquake, including landslides, slope

collapses, and debris flows, in 16 hard-hit counties in Sichuan Province, China, through field surveys and remote sensing interpretation. The traditional method is more accurate but relies too much on expert experience and requires significant time and effort. The pixel- or object-based feature thresholding method sets one or more thresholds for landslide identification by statistically analyzing a specific landslide area's spectral, textural, or geomorphological features [5,6]. However, the threshold method often sets the feature threshold according to the specific research area, which has a small scope of application and weak generalization. When the research area changes, the detection effect is not good. The change detection method is based on two-dimensional optical images or three-dimensional topographic data for two or more periods of remote sensing data of the same location for landslide area change detection [7–10]. Change detection is better for applying fresh landslides but requires time series remote sensing data. The most widely used are optical image-based machine learning detection methods, such as support vector machine (SVM) [11,12], random forest (RF) [13,14], and artificial neural network (ANN) [15–17]. However, such methods require extracting a lot of image feature data and conducting many experiments on feature selection and hyperparameter debugging, which is more workload. In recent years, deep learning technology has developed rapidly, especially Convolutional neural networks(CNN)-based detection algorithms, which have made a series of achievements in the field of image processing [18–23], including image classification, object detection, and semantic segmentation, which can automatically extract multi-layer features from the original image and process complex images effectively [24,25]. Ghorbanzadeh et al. [26] used ANN, SVM, RF, and CNN methods to detect landslides in the southern part of Rasuwa district, Nepal. The results show that CNN-based detection methods require less manual supervision and can be easily applied to other regions by simply retraining the model with other relevant training data. According to the image detection process, CNN-based target detection algorithms are divided into one-stage and two-stage detection methods. The one-stage algorithms of the SSD [27] YOLO series [28–32] belong to end-to-end detection and are characterized by higher speed. The two-stage algorithms represented by Faster R-CNN [33] have slightly lower detection speeds but higher detection accuracy. The above two methods have become the hotspots of landslide detection research to realize the automatic detection of landslides in a certain area [34,35]. Yuanzhen et al. [35] used three target detection algorithms, Mask R-CNN, RetinaNet, and YOLO v3, to automatically identify ancient loess landslides in the Loess Plateau of China. The results show that the two-stage target detection algorithm (Mask R-CNN) is more effective and suitable for detecting old loess landslides. Libo et al. [34] designed a lightweight and fast YOLO-SA algorithm to detect potential landslide areas in Qiaojia and Ludian counties of Yunnan Province, China, with a model F1 score of 90.65%, a miss detection rate of 1.56%, an error rate of 16%, and a detection speed of 42 f/s. Yet, the detection accuracy of the model needs to be improved due to the small landslide dataset and human marking errors. However, most of the existing studies are aimed at landslide detection in specific regions [35–38], with low accuracy and poor robustness for landslide detection in multiple categories in different terrain and landscapes, which has not been widely used.

To solve the above problems, this paper proposes a universal method (YOLOX-Pro) based on improved YOLOX [32] to detect landslides in optical remote sensing images. This method improves the detection accuracy for landslides in different geomorphological environments and has excellent detection capability for small and complex landslides. The main contributions of this work are as follows. First, a mixed historical landslide sample dataset was established. Then, the varifocal loss function was used to solve the miss detection and poor accuracy for small landslides [39], and the attention mechanism was introduced to improve the identification ability for landslide areas [40–42]. In addition, four object detection models were compared with YOLOX-Pro while testing the effect of three attention mechanisms on YOLOX-Pro detection performance. Finally, the effect of the location and quantity of attention mechanism on the YOLOX-Pro were compared, and the

detection ability of the model for UAV landslide images was tested. The proposed method has strong generalization and practical performance, with broad application prospects.

## 2. Study Area and Dataset

In this study, thirty-eight study areas in China where landslides have occurred were selected for research. The study areas include the western alpine valley and the hilly eastern area, and the triggering factors of landslides include earthquakes, precipitation, and human activities. The landslide types cover earth slides, rockslides, and a few debris flows. Within each selected study area, there are multiple landslide points distributed as shown in Figure 1.

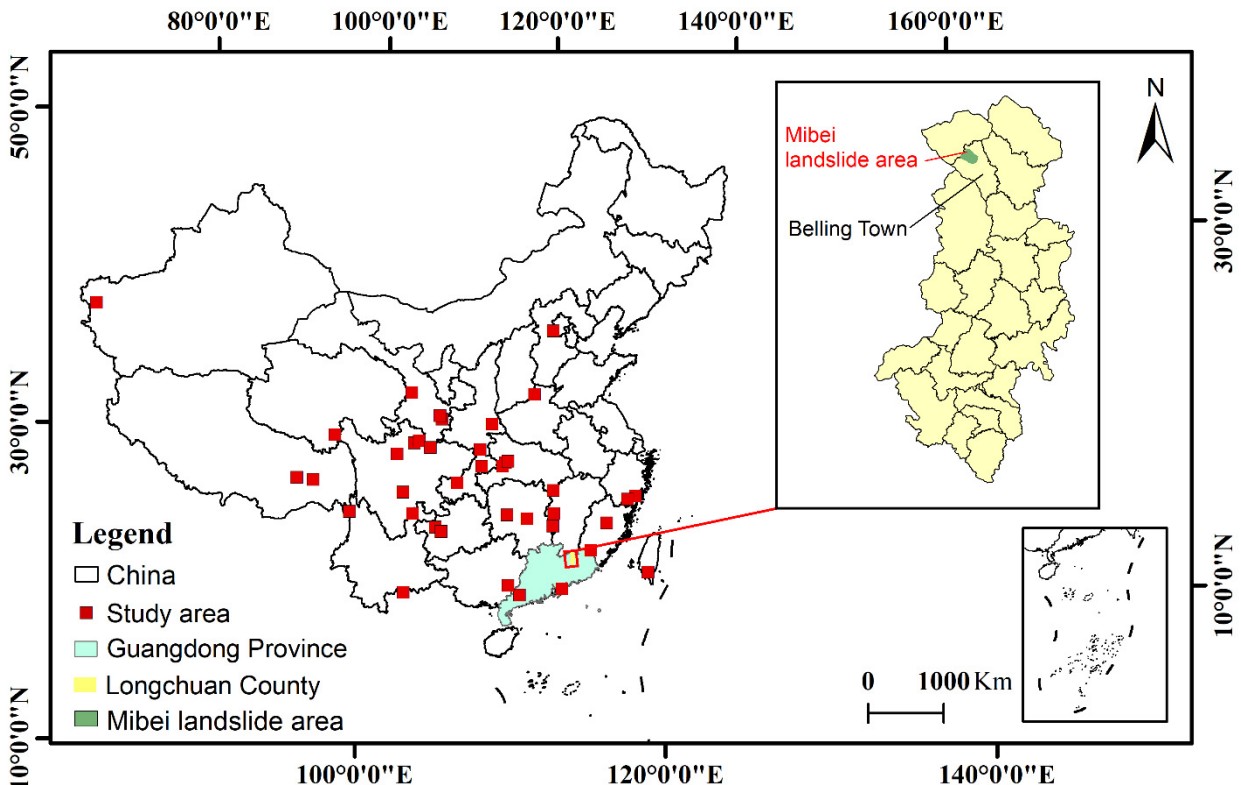

**Figure 1.** The location of the study area. The red rectangular boxes are landslide study areas, with multiple landslides distributed within each area.

We thoroughly considered the diversity of landslide types and distribution based on thirty-eight study areas and extracted these areas' optical remote sensing images using Google Earth software. We selected 1200 images containing landslides as landslide samples and 1200 images not containing landslides as negative samples. Finally, the above study samples were visually interpreted by experts to ensure the accuracy of the image categories. The landslide samples consist of 720 earth slides, 400 rockslides, and 80 debris flows. Most of the images in this dataset have a spatial resolution of 1 m or 0.5 m, with image dates ranging from 2007 to 2021. The negative samples are used to increase the model's ability to discriminate landslides, including mountains, hills, farmland, rivers, roads, and cities. The above two samples formed the CN dataset for this study, which was used to train and test the research model of this paper.

Mibei Village is located in Beiling Town, Longchuan County, Heyuan City, Guangdong Province, China. The heavy rainfall that lasted four days, from 10 June 2019, caused hundreds of small landslides in the area, distributed near roads, houses, and gullies [43]. We obtained Google Earth optical remote sensing images of the area in 2020 with a spatial

resolution of 0.60 m, including 267 landslides (Figure 2), for additional testing of the model's performance.

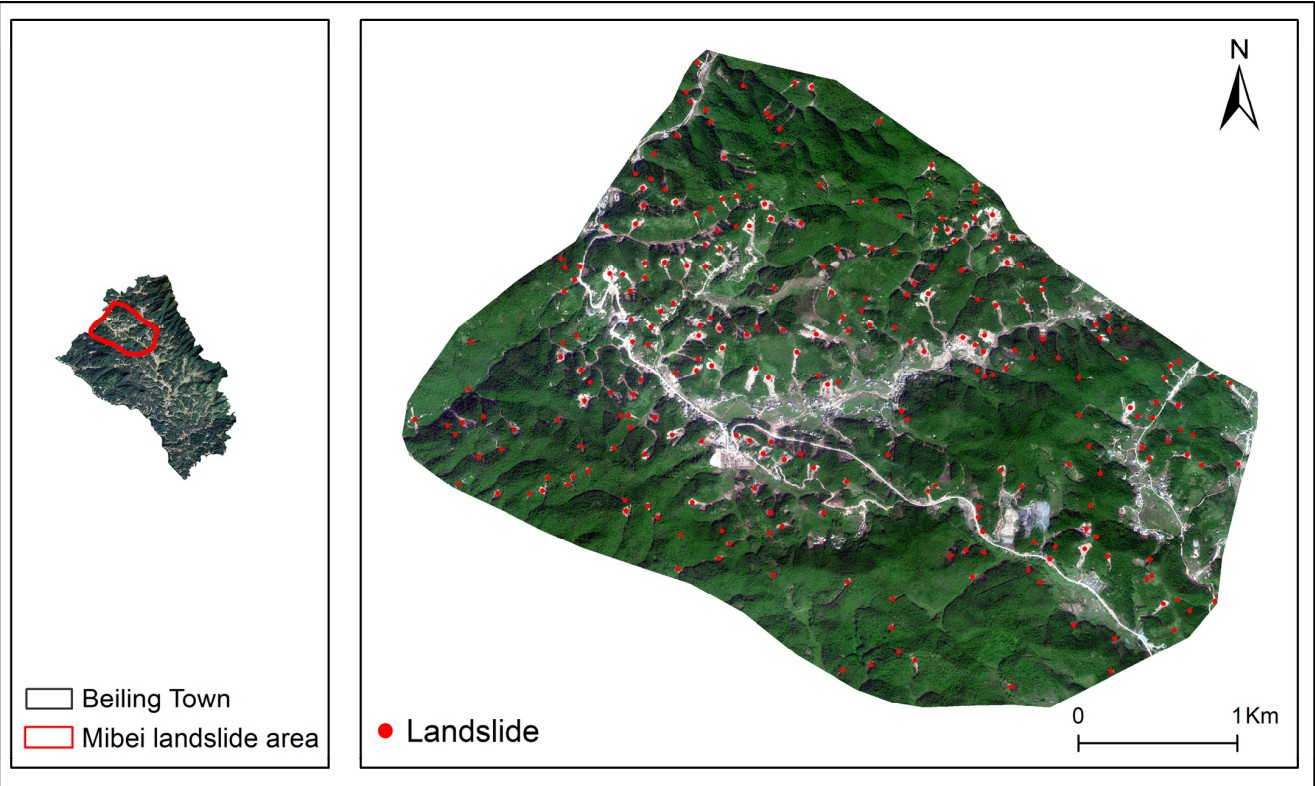

**Figure 2.** The landslide area is located in Mibei Village, Beiling Town, Longchuan County, Heyuan City, Guangdong Province, China. The red points are the locations of landslides identified after the survey.

The UAV (Unmanned Aerial Vehicle) dataset consists of 750 UAV images, including 250 landslide samples and 500 non-landslide negative samples, also visually interpreted by experts. One hundred and fifty landslide images and negative images were collected from the Internet, and another 100 landslide images were obtained from the literature [44]. The UAV dataset was used to evaluate the model's portable performance, and the landslides' time and location were unknown.

The CN and UAV datasets are composed as shown in Table 1, and the research sample for this paper is shown in Figure 3.

**Table 1.** The dataset in this paper. Each sample is an image, the landslide sample contains one to more landslides, and there are no landslides in the negative sample.

| Dataset | Images | | Total Images |
|---|---|---|---|
| CN dataset | 1200 landslides samples | 720 earth slides<br>400 rock slides<br>80 debris flows | 2400 |
| | 1200 negative samples | | |
| UAV dataset | 250 landslides samples | | 750 |
| | 500 negative samples | | |

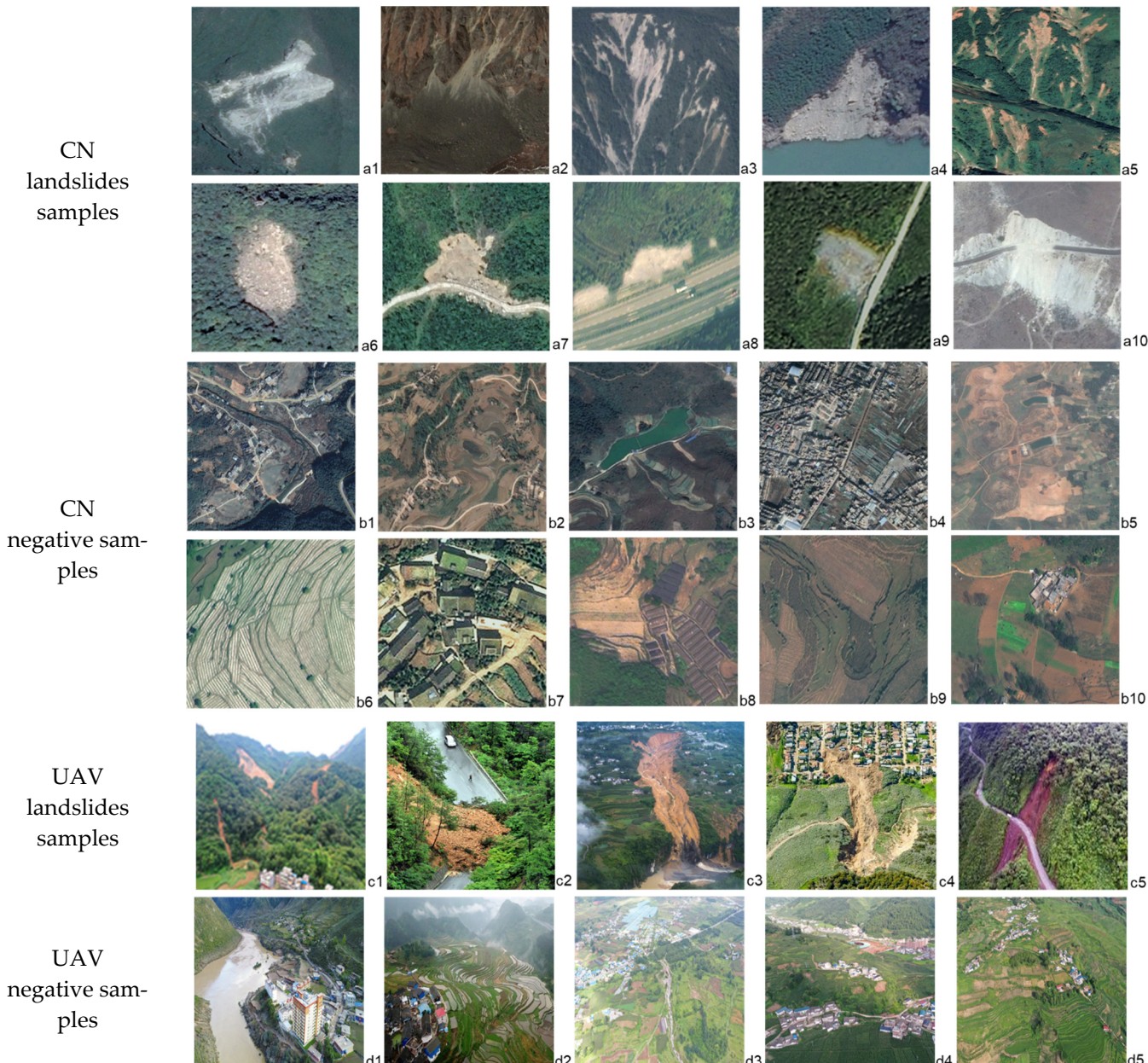

CN landslides samples

CN negative samples

UAV landslides samples

UAV negative samples

**Figure 3.** The research sample for this paper. The landslide samples contain one or more landslides per image, and the negative samples are environmental images that do not contain landslides. CN landslides samples: (**a1**–**a10**); CN negative samples: (**b1**–**b10**); UAV landslides samples: (**c1**–**c5**); UAV negative samples: (**d1**–**d5**).

## 3. Methods

This study constructed a YOLOX-Pro model based on the improved YOLOX [32] model and proposed a method applied to complex hybrid landslide detection. First, the principles of CNN and YOLOX algorithms were briefly reviewed. Then, the algorithm was investigated in two aspects to make it applicable to landslide detection tasks: (1) Replaced the loss function to solve the uneven distribution of landslide samples and improve the model's ability to detect dense and small landslides and complex landslides, thus improving the detection Recall. (2) Based on the YOLOX, the attention mechanism was used to enhance the algorithm's ability to identify landslides in the environment and thus improve the detection accuracy forming the YOLOX-Pro model. Moreover, the Common Objects

in Context (COCO) [45] evaluation metrics were introduced to compare the detection performance of various models in detail.

### 3.1. Convolutional Neural Network

CNN is widely used in image recognition tasks. Its basic structure consists of a convolutional layer, an activation function, a pooling layer, and a fully connected layer, as shown in Figure 4. The typical convolution operation is as follows:

$$z^l = \sigma\left(w^l \times a^{l-1} + b^l\right) \tag{1}$$

where $z^l$ is the output feature map of the lth layer, $w^l$ and $b^l$ is the convolution kernel and bias of the lth layer, respectively, $a^{l-1}$ is the input image or feature map, and $\sigma$ represents the activation function. The convolution layer is equivalent to a feature extractor, using different sizes of convolution kernels to extract features at different scales of the image, e.g., c $\times$ 3 $\times$ 3, where c is the number of input channels, 3 $\times$ 3 denotes the size of the extracted area, and the output is called a feature map. Each feature map is used as a class of extracted image features. Multiple convolutional kernels are usually used to obtain different feature maps to improve the representation capability of the convolutional network. The activation function is a nonlinear function, such as sigmoid $\left(1/\left(1+e^{(-x)}\right)\right)$ or SiLU $(x \times \text{sigmoid}(x))$, which aims to enhance the representation and learning ability of the neural network. The pooling layer is usually used for feature selection, reducing the feature map size and the number of parameters, with maximum pooling and average pooling being the most used. Furthermore, this study also uses average pooling along with the *X*-axis and *Y*-axis directions, as shown in Figure 5. The fully connected layer at the tail of the network maps the high-dimensional features to the low-dimensional space and classifies the extracted features.

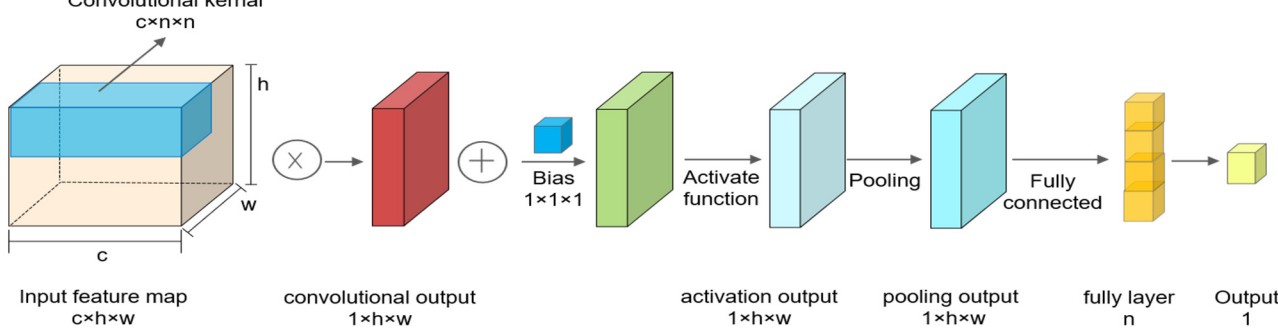

**Figure 4.** The basic CNN structure. The "$\times$" and "+" with circles are multiplication and addition operations.

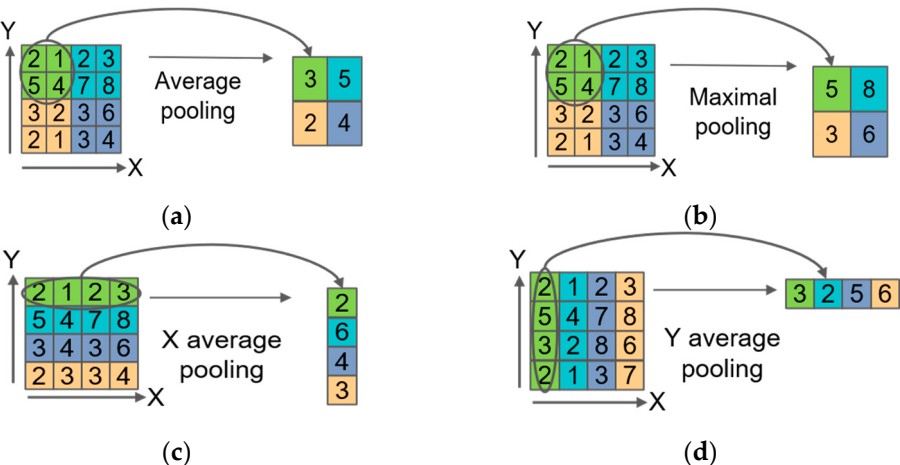

**Figure 5.** Four typical pooling operations: (**a**) Average pooling; (**b**) Maximal pooling; (**c**) X average pooling: one-dimensional features are encoded along the horizontal direction and then aggregated along the vertical direction; (**d**) Y average pooling: one-dimensional features are encoded along the vertical direction and then aggregated along the horizontal direction.

### 3.2. YOLOX Algorithm

YOLOX is a one-stage target detection algorithm proposed by Megvii Technology, China, in 2021, improving the basic network structure of YOLOV3-SPP [29]. Compared with other object detection algorithms of the YOLO series, the standard network structure of YOLOX has significantly improved speed and accuracy. For lightweight network structure, it has high detection accuracy with fewer parameters. The YOLOX model consists of three modules: Backbone, Neck, and Head. Its network structure is shown in Figure 6.

As shown in Figure 6, the Focus [31], CBS, Cross Stage Partial Network (CSP) [46], and Spatial Pyramid Pooling (SPP) [47] are the basic modules of YOLOX, which are used to extract and transform the input features. The CBS module consists of Convolution, Batch Normalization (BN) [48], and SILU functions. The Bottleneck is a structural unit in ResNet [49] that is used to improve the nonlinearity of the network structure and reduce the computational effort.

Backbone extracts feature from the input images, and the obtained feature sets are called feature map. A total of three effective feature maps containing different scale information are received in the backbone network, which is 1/8, 1/16, and 1/32 of the input image size, respectively.

In the Neck, the effective feature maps are fused with up-sampled and down-sampled features to get three enhanced feature maps containing richer picture information, each of which is a collection of a large number of feature points.

The Head is the classifier and regressor of YOLOX, which detects the feature map output from the Neck part, determines whether the feature point corresponds to the object, and realizes the classification and detection of the picture. As an example, the detection images contain eighty categories of objects. YOLOX uses three Decoupled Heads to detect feature maps of size $20 \times 20 \times (80 + 1 + 4)$, $40 \times 40 \times (80 + 1 + 4)$, and $80 \times 80 \times (80 + 1 + 4)$, corresponding to eighty categories, one confidence score, and four regression parameters information, respectively. Among the four regression parameters, the first two are the offsets of the center point of the prediction box relative to the corresponding feature points, and the last two are the parameters of the width and height of the prediction box relative to the logarithmic index. After some Concatenation and Reshaping operations, a set of feature vectors (85,8400) containing the above information is output. Among them, 8400 refers to the number of prediction points, (corresponding prediction box size $8 \times 8$, $16 \times 16$, $32 \times 32$), and eighty-five refers to the information of the prediction point (Cls, Reg, Obj). Finally, the prediction points are matched with the objects in the images, and the

best prediction boxes are selected as the detection results after confidence ranking and Non-Maximum Suppression (NMS) [50] (Figure 7).

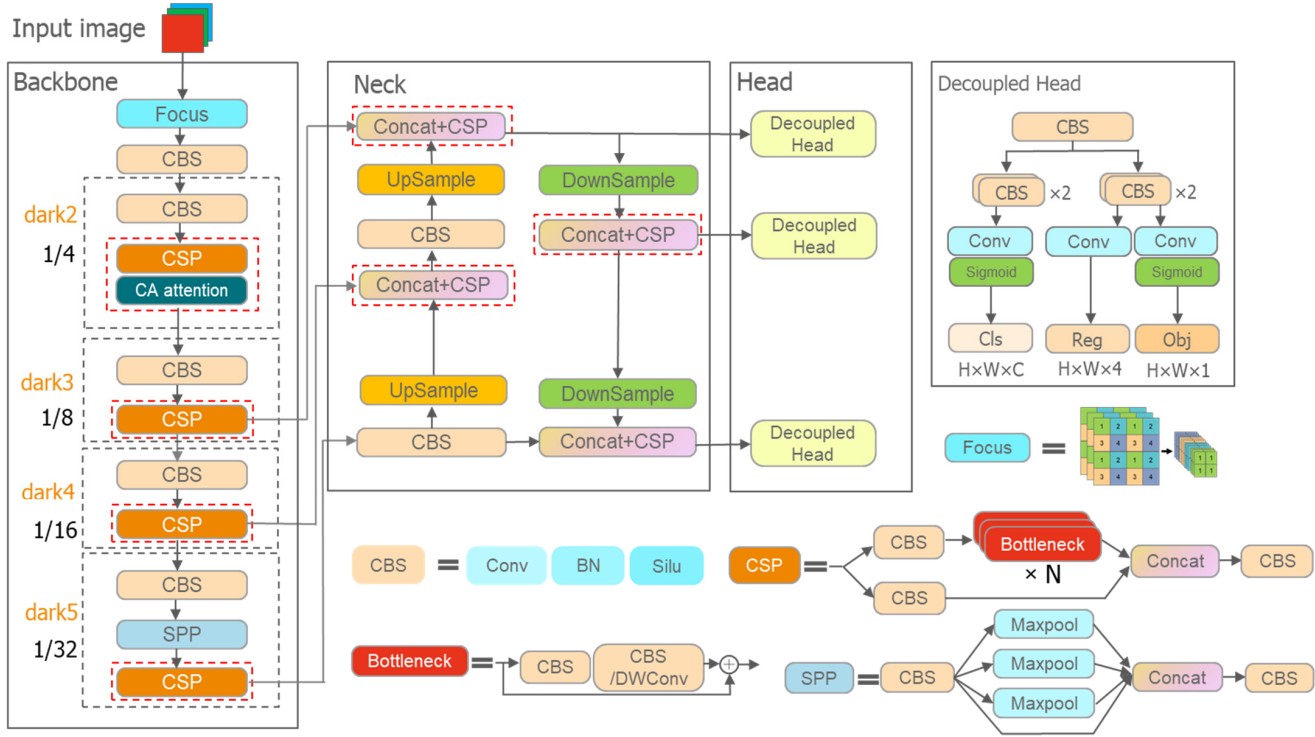

**Figure 6.** YOLOX network structure. The red dotted boxes are the alternative locations for adding the attention module. The Conv, BN, and Silu denote Convolution, Batch Normalization, and SILU, respectively. The Concat and Maxpool denote Concatenation and Maximum pooling operations. The DWConv is a variant of the Convolution operation. The Cls, Reg, and Obj represent the classification, the regression parameters of each feature point, and the detected target's confidence score information from the input image. The prediction box can be obtained by adjusting the regression parameters. The confidence score represents the probability that each feature point prediction box contains an object. The H, W, and X indicate the feature maps' width, height, and number of channels.

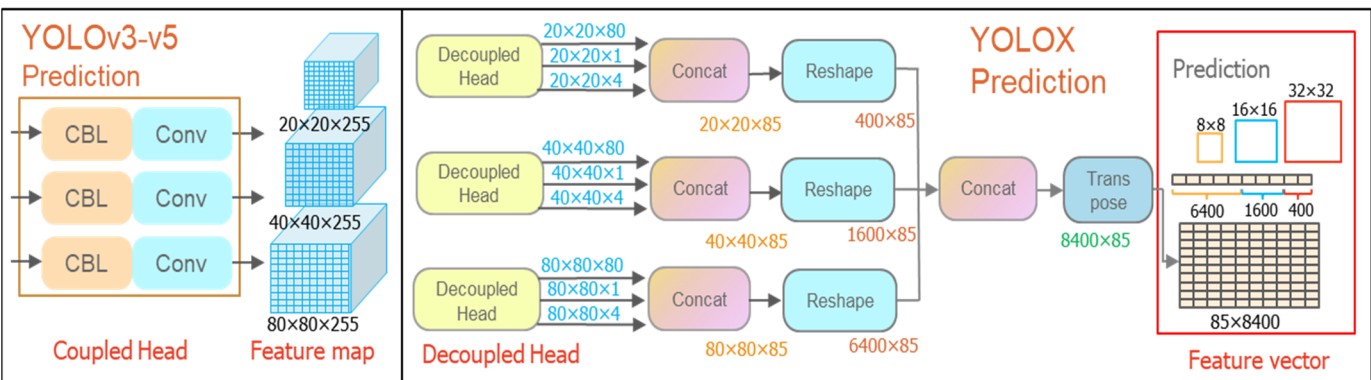

**Figure 7.** YOLO series detection head structure. The CBL module consists of Convolution, Batch Normalization, and LeakyRelu functions $(x, x > 0; a \times x, x \leq 0)$ to extract and transform the input features, where $a$ is a tiny constant. The number of prediction boxes corresponding to the three types of feature points is 6400, 1600, and 400, respectively.

The main improvements of the YOLOX network are as follows.

Yolv3-v5 uses coupled heads [51,52] to achieve classification and regression tasks using $1 \times 1$ convolution, which limits the expressiveness of the detected images. In contrast, YOLOX uses decoupled heads that perform classification and regression separately and are integrated at the final prediction, improving the model's detection accuracy and convergence speed. The structure of YOLO series detection heads is shown in Figure 7.

YOLOX uses the Anchor Free detection method [53,54], which outputs only one set of feature vectors and detects about 700,000 feature points. In contrast, YOLOv3-v5 uses the Anchor Based method, which outputs three sets of feature maps and requires about 2 million feature points to predict, and YOLOX reduces about 2/3 of the parameters (Figure 7). Moreover, YOLOX adopts the SimOTA [51] label assignment strategy of dynamically matching positive samples for targets of different sizes, giving the model a certain adaptive ability to match the detected targets dynamically. During the training process of the neural network, the prediction results become more and more accurate as the number of iterations increases and the model parameters are updated.

As shown in Figure 6, the Focus [31], CBS, Cross Stage Partial Network (CSP) [46], and Spatial Pyramid Pooling (SPP) [47] are the basic modules of YOLOX, which are used to extract and transform the input features, where *a* is a tiny constant. The number of prediction boxes corresponding to the three types of feature points is 6400, 1600, and 400, respectively.

### 3.3. YOLOX-Pro Algorithm

We improved the YOLOX algorithm to meet the requirements of landslide detection tasks and named the new algorithm YOLOX-Pro (Figure 8). The YOLOX-Pro model includes five network sizes: Nano, Tiny, S, M, and L, with increasing parameters and computational power, as well as progressively increasing detection performance (Table 2). A network of appropriate size can be selected for landslide detection according to the detection accuracy requirements and the capability of computing equipment. The improvement of the YOLOX-Pro model is mainly in two aspects. First, the classification loss function of YOLOX was replaced by the VariFocal loss function, which is used as the base model for the study. Second, three attention modules, Squeeze-and-Excitation Module (SE) [40], Convolutional Block Attention Module (CBAM) [41], and Coordinate Attention (CA) [42], were added to the base model for comparison, and the optimal model was selected to form the YOLOX-Pro model.

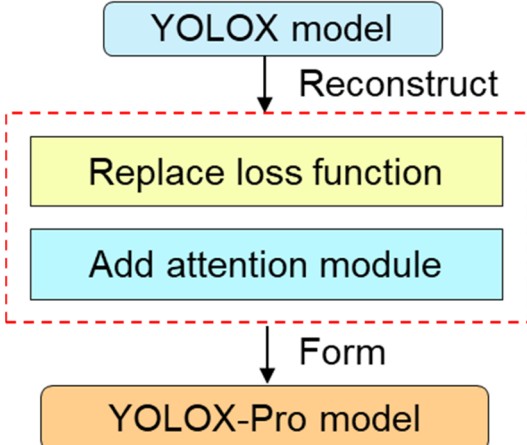

**Figure 8.** Model design flowchart.

**Table 2.** The Parameters of YOLOX and YOLOX-Pro.

| | Parameters (Mb) | | | | |
|---|---|---|---|---|---|
| Network Size | Nano | Tiny | S | M | L |
| YOLOX | 0.91 | 5.06 | 9.00 | 25.30 | 54.20 |
| YOLOX-Pro | 1.42 | 6.51 | 10.30 | 26.50 | 57.51 |

### 3.3.1. Reconstructing the Loss Function

The loss function is usually expressed using $L(p, y)$, a non-negative real-valued function to estimate the degree of difference between the predicted value $p$ and the label $y$ (true value). The smaller the loss function, the better the model's prediction is within a certain range. In binary classification tasks where the output $y$ has only two discrete values of $[0, 1]$, a binary cross-entropy loss function is usually used to discriminate. Where $p$ is the probability that the predicted outcome is a positive sample, $(1 - p)$ denotes the negative sample probability, and $y$ is the true label of the sample, with a positive sample label of 1 and a negative sample label of 0. The basic formula is as follows.

$$L(p, y) = -y \log(p) - (1 - y) \log(l - p) \tag{2}$$

There are three types of losses in YOLOX: The prediction box position loss, which is used to fit the bounding box during the training process gradually. The target score loss reflects the actual probability of the predicted outcome. The target category score loss reflects the probability that the predicted target is a certain category.

High-resolution optical remote sensing images are usually used in landslide detection tasks, while landslides occupying a tiny percentage of the image can create a severe proportional imbalance problem. In addition, difficult samples (small landslides and complex landslide objects that are difficult to identify) can pose a significant challenge to the detection task. Therefore, the VariFocal loss function is used instead of the target score loss function in the original model to solve the above problem [39]. VariFocal loss function improves the recognition of different types of targets by setting hyperparameters to reduce the negative sample loss weights while increasing the hard sample weights. The VariFocal loss function is designed based on the binary cross-entropy loss function and is expressed as follows:

$$VFL(p, q) = \begin{cases} -q(q \log(p)) + (1 - q) \log(l - p) & q > 0 \\ -\alpha p^\gamma \log(1 - p) & q = 0 \end{cases} \tag{3}$$

where $p$ is the predicted IoU-aware classification score (IACS), and $q$ is the target score, the intersection ratio IoU (Intersection over Union) is used to measure the degree of resynthesis of the prediction box and the ground truth. The larger the value indicates, the more accurate positioning of the prediction boxes, taking the value range [0:1]. The IoU is expressed as follows:

$$IoU = \frac{area\left(B_p \cap B_{gt}\right)}{area\left(B_p \cup B_{gt}\right)} \tag{4}$$

where $B_p$ is the prediction box, and $B_{gt}$ is the ground truth (actual box).

Positive samples, $q$ are set to the IoU between the prediction box and the ground truth, and all negative samples have a $q$ value of 0. The parameters $\alpha$ and $\gamma$ are used to balance the losses of positive and negative samples. The negative samples are multiplied by $p$ to reduce their loss weights, due to the small number of positive samples. Positive samples are multiplied by $q$ to increase the loss weights because positive samples with higher IoU values have larger loss values. The losses can be focused on those high-quality samples during training. Finally, the overall positive and negative samples are weighted using $\alpha$ to balance the losses.

### 3.3.2. Add Attention Module

In computer vision, the attention mechanism aims to keep the algorithm focused on key information and suppress redundant information. We added the attention mechanism to the base model to improve the algorithm's recognition of landslide areas and thus improve the detection performance. Three attention modules, SE, CBAM, and CA, were added to the base model dark2 block after the CSP component, as shown in Figure 6.

Attention Mechanism

The attention mechanism commonly used in convolutional neural networks can be categorized depending on the domain of attention:

Spatial Domain: Compress the channels of the images to get the key information containing only spatial dimensions, and then use the key information to produce weights, which in turn act on the spatial data for the generation and scoring of masks. The representative Spatial Attention Module [52].

Channel Domain: Compressing the spatial dimension of the image to get only the key information of the channel dimension, using the extracted key information to produce weights, and then reacting to the channel data to generate and score the mask. The representatives are Squeeze-and-Excitation Module (SeNet) and the Efficient Channel Attention Module (ECA-Net) [55].

Hybrid domain: Combining spatial domain and channel domain, focusing on both spatial information and channel information of the image, represented by Convolutional Block Attention Module (CBAM) and Coordinate Attention (CA).

Coordinate Attention Mechanism

Coordinate Attention (CA) is a new lightweight attention mechanism for mobile networks that effectively improves model performance while incurring little computational overhead. CA embeds the location information into the channel attention and then decomposes the channel attention into two one-dimensional feature encoding processes (Figure 5c,d). Then, the above two features are aggregated along two spatial directions separately to obtain the remote dependencies in one spatial direction while preserving the precise location information in the other. The generated feature maps are encoded into direction-aware and location-aware attentional feature maps, which can be applied to the input feature maps in a complementary way, enabling the network to locate the object of interest more accurately. The CA workflow is shown in Figure 9.

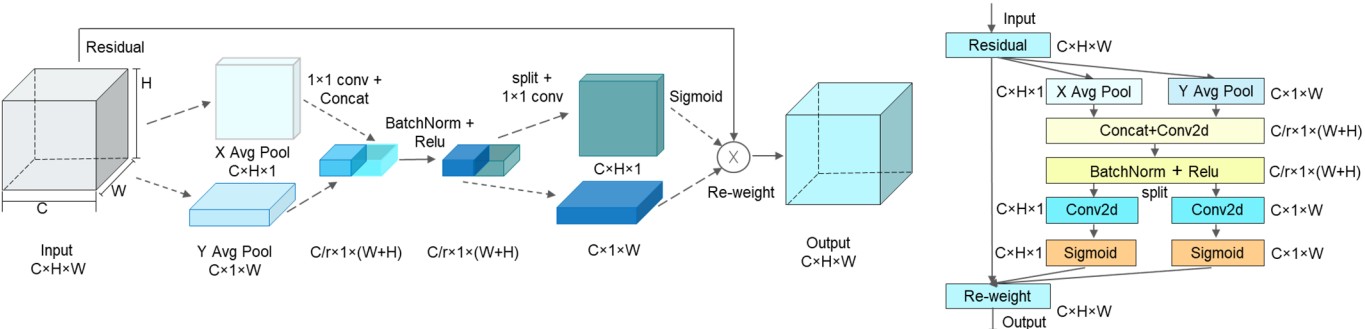

**Figure 9.** The Coordinate Attention workflow. C, H, and W are the number of channels, height, and width of the feature map, respectively. The r denotes the compression ratio of the channels.

The CA channel attention module encodes channel attention blocks' relationships and remote dependencies through two steps: embedding coordinate information and generating coordinate attention. The embedding of coordinate information is a global average convergence operation along with the horizontal and vertical directions for a given input $F_{in} \in R^{1 \times H \times W}$, to obtain the location information associated with the one-

dimensional feature in the horizontal direction (Figure 5c) and the one-dimensional feature in the vertical direction (Figure 5d), and is expressed as follows:

$$f_h^c(h) = \frac{1}{W} \sum_{0 \le i < W} F_{in}^c(h, i) \tag{5}$$

$$f_w^c(w) = \frac{1}{H} \sum_{0 \le j < H} F_{in}^c(j, w) \tag{6}$$

In Equations (5) and (6), at channel C, the input $F_{in}^C \in R^{1 \times H \times W}$ with height $H$ and width $W$, encode along the horizontal direction to get $f_h^c \in R^{1 \times H \times 1}$, and along the vertical direction to get $f_w^c \in R^{1 \times 1 \times w}$.

In the generation step of the coordinate attention, all channels of $f_h^c$ and $f_w^c$ are connected together to obtain $f_h \in R^{C \times H \times 1}$ and $f_w \in R^{C \times 1 \times W}$. Then, they are concatenated and send to a shared $1 \times 1$ Convolutional transformation function $F^{1 \times 1}$ for the compression transformation of the channel dimension; the hyperparameter r controls the compression ratio, $[\cdot, \cdot]$ is the concatenation operation along the horizontal and vertical features, and $\delta$ represents the Relu $(\max(0, x))$ function here, and finally, the batch normalization operation $(BN)$ is performed to obtain $f_{hw} \in R^{C/r \times (H+W)}$. The representation is as follows:

$$f_{hw} = BN(\delta(F^{1 \times 1}([f_h, f_w]))) \tag{7}$$

Then, split $f_{hw}$ along the spatial dimension into two separate tensors $f'_h \in R^{C/r \times H}$ and $f'_w \in R^{C/r \times W}$, followed by two $1 \times 1$ convolutional transformations $F_h^{1 \times 1}$ and $F_w^{1 \times 1}$, next multiplied by the sigmoid function $\sigma$ to obtain the attention weights.

Finally, the horizontal and vertical attention weights are dimensionally expanded to $R^{C \times H \times W}$, and multiplied with the input data $F_{in}$ to obtain the coordinate attention output feature map $F_{out} \in R^{C \times H \times W}$, which can be formulated as:

$$f_h'' = \sigma\left(F_h^{1 \times 1}\left(f_h'\right)\right) \tag{8}$$

$$f_w'' = \sigma\left(F_w^{1 \times 1}\left(f_w'\right)\right) \tag{9}$$

$$F_{out} = F_{in}(i, j) \times f_h''(i) \times f_w''(j) \tag{10}$$

The coordinate attention module applies horizontal and vertical attention weights to the input feature map to more accurately locate the exact position of the object of interest, thus helping the entire model better identify objects.

### 3.4. Model Evaluation Methods

The confusion matrix-based method is commonly used to evaluate the performance of the object detection model, but it can only evaluate the model's overall performance. This study introduces the COCO detection evaluation metrics [45], which consist of 12 metrics to evaluate the performance of the target detection model in detail. The COCO metrics are defined based on the confusion matrix, which is explained in detail below.

The confusion matrix evaluation metrics include precision (P), recall (R), and average precision (AP). Precision indicates the probability of being correct among the targets detected as positive samples. Recall indicates the probability of being correctly identified among all positive samples. AP is a comprehensive evaluation index calculated by the area under the Precision and Recall curves. Higher AP means better model performance. The definitions are as follows:

$$Precision = \frac{TP}{TP + FP} \tag{11}$$

$$Recall = \frac{TP}{TP + FN} \tag{12}$$

$$AP = \int_0^1 P(R)dR \tag{13}$$

$$\text{IoU}(pbox, gt) = \frac{area(pbox \cup gt)}{area(pbox \cap gt)} \begin{pmatrix} pbox : predictionbox \\ gt : groundtruth \end{pmatrix} \tag{14}$$

In the above equation, the IoU (intersection over union) between the prediction box and the ground truth is used to determine the class of the prediction result, and the most common IoU threshold is 0.5. If the IoU is greater than 0.5, the prediction box is labeled as true positive (TP), else the prediction box is labeled as false positive (FP). False negative (FN) is missed detection. In this study, TP, FP, and FN represent the number of correctly, incorrectly, and missed detected landslides.

The COCO evaluation metrics (Figure 10). AP is a primary challenge metric and is the average AP value when the IoU threshold increases from 0.5 to 0.95, with the step size being 0.05. AP0.5 and AP0.75 were the AP value when the IoU threshold was separately set to 0.5 and 0.75. AP0.75 is a strict evaluation metric.

**Average Precision (AP):**
- AP　% AP at IoU=0.50: 0.05: 0.95 (**primary challenge metric**)
- $AP^{0.5}$　% AP at IoU=0.50 (PASCAL VOC metric )
- $AP^{0.75}$　% AP at IoU=0.75 (strict metric )

**AR Across Scales:**
- $AP^{small}$　% AP for small objects: area < $32^2$
- $AP^{medium}$　% AP for medium objects: $32^2$ < area < $96^2$
- $AP^{large}$　% AP for large objects: area > $96^2$

**Average Recall (AR):**
- $AR^1$　% AR given 1 detection per image
- $AR^{10}$　% AR given 10 detection per image
- $AR^{100}$　% AR given 100 detection per image

**AP Across Scales:**
- $AR^{small}$　% AR for small objects: area < $32^2$
- $AR^{medium}$　% AR for medium objects: $32^2$ < area < $96^2$
- $AR^{large}$　% AR for large objects: area > $96^2$

**Figure 10.** The detection evaluation metrics of COCO.

The average recall (AR) is the maximum recall given a fixed number of detections per image, averaged over categories and IoUs, and the IoU value is set to 0.5 in this study. AR is divided into two categories. The maximum number of targets in the detected objects is divided into AR1, AR10, and AR100, reflecting the model's detection performance for multi-target objects. The pixel area is divided into APsmall, APmedium, and ARmax, reflecting the detection performance of the model for targets of different scales, where the pixel area less than $32^2$ was marked as the small target, and the area between $32^2$ and $96^2$ was marked as the medium target. The rest was marked as the large target. The AR formula is as follows.

$$AR = \frac{Recall}{n} \tag{15}$$

The units of the above evaluation metrics are all in percentages. Some of the above metrics were selected to comprehensively evaluate the models used in this study. AP0.5, Recall, and Precision metrics were selected to evaluate the overall detection performance of the model. AP, AP0.75 are more strict evaluation metrics for comparing subtle differences in model performance. APsmall, Arsmall, and AR10 were used to evaluate the model's detection performance for minor landslides and cluster landslides.

## 4. Experimental Setup and Results

In this section, we first described the setup of the experiment. Then, we added three attention modules to the base model for comparison and selected the best model to form the YOLOX-Pro model. Next, the YOLOX-Pro was compared with other models and analyzed

in detail. Finally, we present and analyze the landslide detection results of the YOLOX-Pro model, including the CN test set, the Mibei area, and the UAV landslide images.

### 4.1. Experimental Setup

The CN dataset was randomly divided into training, validation, and test sets with a ratio of 6:2:2, and the number of images is 1440, 480, and 480, respectively (Table 3). Then, the image sizes of the training and validation sets were resampled to 640 × 640 pixels, and the rectangular bounding boxes and labels of the landslides were labeled using the LabelImg software. Landslide experts verified the annotation results to ensure the accuracy of the annotation. The training set was utilized for training the model, the validation set was employed to select the optimal model, and the test set was not processed and used to evaluate the model's performance (Figure 11).

**Table 3.** The details of dataset division. The UAV dataset does not have validation set. The Mibei landslide area is only divided into test set for landslide detection.

| Data Composition | CN Dataset | | UAV Dataset | | Mibei Landslide Area |
|---|---|---|---|---|---|
| | Landslide images | Negative images | Landslide images | Negative images | images |
| Train | 770 | 670 | 150 | 400 | - |
| Val | 180 | 300 | - | - | - |
| Test | 250 | 230 | 100 | 100 | 26 |
| Total | 1200 | 1200 | 250 | 500 | 26 |

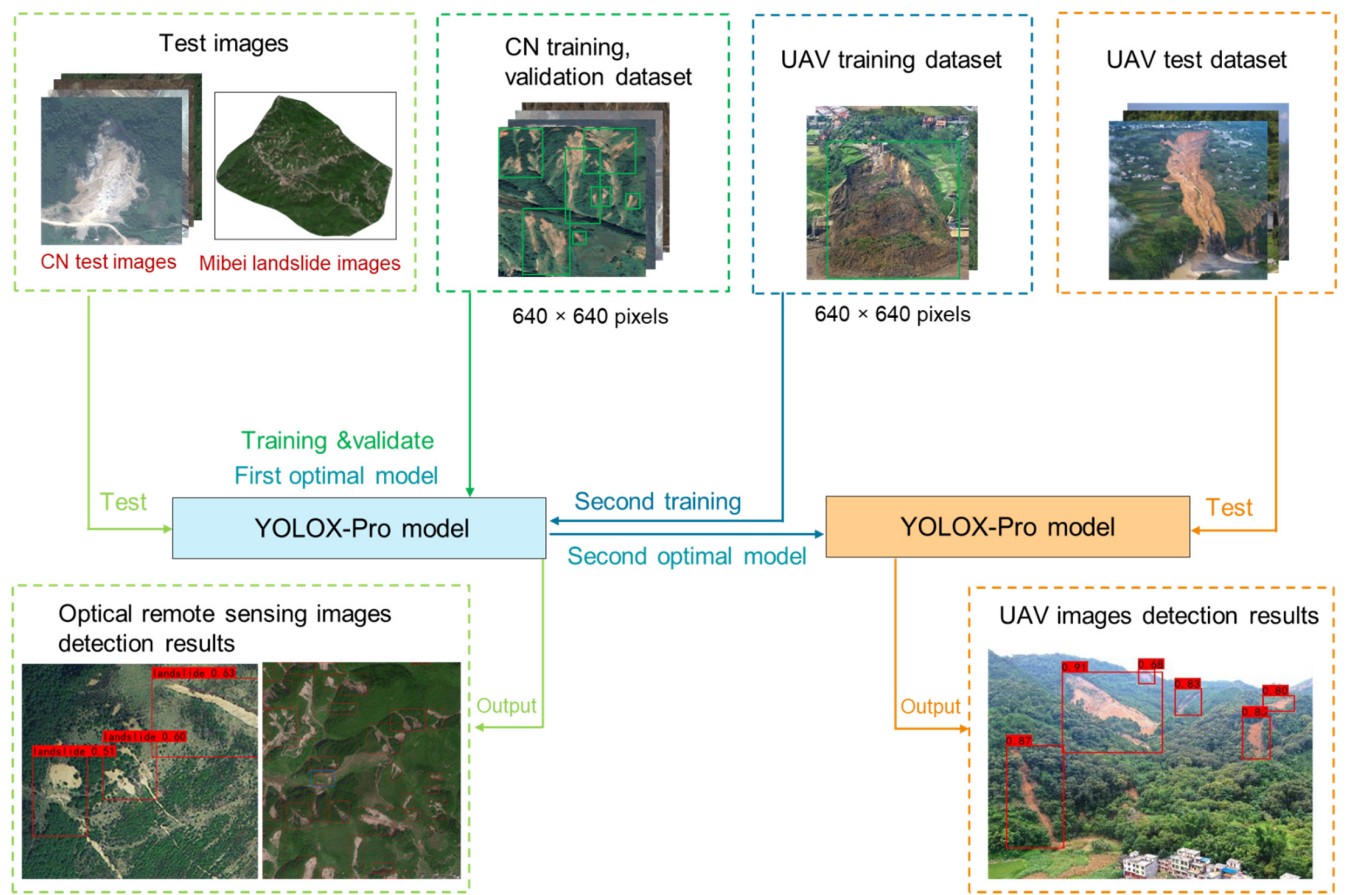

**Figure 11.** The workflow of landslide detection from CN dataset, UAV dataset, and Mibei landslide area.

The optical remote sensing image of the Mibei landslide area was cropped into 1920 × 1920 pixels patches from the top left corner. Twenty-six images were obtained and fed to the optimal model for additional landslide detection. The overlap size of each

crop direction is 300 pixels to ensure that each landslide has a complete image for detection. (Figure 11).

The UAV dataset was divided into training and test set, consisting of 500 and 250 images, respectively. Due to the small number of samples, there is no validation set. The training images were resampled to $640 \times 640$ pixels, and the test images were not processed. First, the UAV training images were fed into the first optimal model for the second training. Then, the UAV test images were fed into the second optimal model for testing (Figure 11). The detailed information of the dataset is shown in Table 3.

The experiments were conducted on a workstation equipped with an Intel I CoITM) i7-12700 processor, 32 GB Random Access Memory (RAM), and NVIDIA RTX 3090 graphics processor with 24 GB of video memory, built on the deep learning framework PyTorch and implemented using the Python programming language.

To accelerate and enhance the learning ability of the model, the model was initialized with weights obtained from the ImageNet dataset before training, based on a transfer learning approach [56,57]. The Stochastic Gradient Descent (SGD) [58] optimizer was used for all training processes. The freeze training epochs was 50, and the initial learning rate was $10^{-3}$, and the batch size was 32. The thaw training epochs was 100, and initial learning was $10^{-4}$, and batch size was 16. The weight decay factor is 0.005, with a momentum factor of 0.9. The second experiment was trained for 50 epochs with the same parameter settings as the first.

### 4.2. Model Assessment

The images from the test set were not used for model training and are unknown to the model. Therefore, the trained models are used to perform landslide detection on the test set images to compare the performance of different models.

### 4.2.1. Comparing Different Attention Mechanisms

We added SE, CBAM, and CA attention modules to the base model for comparison to select the most appropriate attention module to form the optimal model. Large-size network YOLOX (m) and small-size network YOLOX (nano) were selected as representatives for the experiment, and the result is shown in Table 4.

**Table 4.** The effect of different attentional mechanisms on the YOLOX-Pro model.

| Model | Params (Mb) | AP (%) | AP0.75 (%) | APsmall (%) | ARsmall (%) |
|---|---|---|---|---|---|
| YOLOX (m) | 25.30 | 47.00 | 48.30 | 32.50 | 46.80 |
| YOLOX-CA (m) | 26.50 | 47.80 | 51.50 | 36.50 | 49.50 |
| YOLOX-CBAM (m) | 26.68 | 47.70 | 50.10 | 35.30 | 47.90 |
| YOLOX-SE (m) | 26.59 | 47.50 | 49.80 | 35.70 | 48.10 |
| YOLOX (nano) | 0.91 | 45.10 | 44.60 | 31.30 | 45.00 |
| YOLOX-CA (nano) | 1.42 | 46.10 | 46.90 | 33.70 | 47.50 |
| YOLOX-CBAM (nano) | 1.44 | 45.70 | 46.70 | 33.10 | 47.20 |
| YOLOX-SE (nano) | 1.43 | 45.60 | 46.60 | 32.80 | 46.90 |

In the CN dataset, the pixel area of landslides in many images is less than $32 \times 32$ pixels corresponding to the actual size of landslides as less than $32^2$ or $16^2$ square meters (Optical remote sensing images of these landslide samples have a resolution of 1 m or 0.5 m). So, the four metrics AP, AP0.75, APsmall, ARsmall, and AR10 are selected to compare the detection performance of different modules in detail.

Table 4 shows that adding the CA module increases the number of parameters less than the CBAM and SE modules compared to the standard YOLOX model. The YOLOX-CA obtains an overall advantage over other models and improves APsmall and ARsmall by more than 2%. In later experiments, we used the YOLOX-CA as the improved model and named YOLOX-Pro.

### 4.2.2. Comparing Different Detection Models

To determine the performance of YOLOX-Pro, a comparison was made with four classical detection models: YOLOX, YOLOv5, Faster-RCNN, and SSD. Three comprehensive metrics, AP0.5, Recall, and Precision, were used to evaluate the detection performance of the above models in our CN dataset.

Table 5 shows that YOLOX-Pro (m) has an obvious advantage over YOLOX (m). AP0.5 reached 84.85%, Recall improved by 6.43%, Precision improved by 3.06%, while the parameters increased slightly by 1.2 Mb. YOLOv5 (m) has the least parameters, but low recall leads to poor performance. The Faster-RCNN has the most parameters and higher Recall but the lowest Precision. The Recall of the SSD is the lowest and performed the worst.

**Table 5.** Comparison of detection performance of different models.

| Model | Params (Mb) | AP0.5 (%) | Recall (%) | Precision (%) |
|---|---|---|---|---|
| YOLOX-Pro (m) | 26.50 | 84.85 | 81.52 | 85.36 |
| YOLOX (m) | 25.30 | 81.73 | 75.09 | 82.30 |
| YOLOv5 (m) | 21.20 | 74.30 | 68.95 | 80.25 |
| Faster R-CNN (resnet-50) | 107.87 | 71.35 | 77.37 | 67.56 |
| SSD (vgg) | 90.07 | 69.15 | 57.29 | 76.19 |

In the CN dataset, the number of landslides in most images is within 10, so the AR10 metrics was added to compare the detection performance of YOLOX and YOLOX-Pro models for images containing multiple landslides.

Compared with standard YOLOX, the YOLOX-Pro has improved all evaluation metrics with a slight parameter increase (Table 6). In particular, the AP0.75 of the S network increased by 5.8%, and the ARsmall of the M network increased by 4.0%. Specifically, the increase of AP for each network of YOLOX-Pro is slight. However, the rise of AP0.75 is apparent, indicating that the model has a greater improved comprehensive detection performance for high-quality objects. The increase of APsmall and ARsmall is significant, and the average rise of AR10 is 2.86%, indicating that the model improves the detection performance of multi-target objects significantly, especially small targets.

**Table 6.** Comparison of detection performance between YOLOX and YOLOX-Pro models.

| | YOLOX | | | | | YOLOX-Pro | | | | |
|---|---|---|---|---|---|---|---|---|---|---|
| | l | m | s | Tiny | Nano | l | m | s | Tiny | Nano |
| Params (Mb) | 54.20 | 25.30 | 9.00 | 5.06 | 0.91 | 57.51 | 26.50 | 10.30 | 6.51 | 1.42 |
| AP (%) | 47.20 | 47.00 | 45.30 | 45.10 | 45.10 | 48.30 | 47.80 | 47.00 | 46.70 | 46.10 |
| AP0.75 (%) | 48.70 | 48.30 | 45.00 | 45.20 | 44.60 | 52.40 | 51.50 | 50.80 | 48.80 | 46.90 |
| APsmall (%) | 33.40 | 32.50 | 32.10 | 31.40 | 31.30 | 37.50 | 36.50 | 35.30 | 34.82 | 33.70 |
| ARsmall (%) | 47.00 | 46.80 | 45.70 | 45.40 | 45.00 | 49.80 | 49.50 | 49.20 | 48.60 | 447.50 |
| AR10 (%) | 58.60 | 58.60 | 57.70 | 57.80 | 57.40 | 61.70 | 61.20 | 60.90 | 60.50 | 60.10 |

To demonstrate the difference in detection performance between YOLOX and YOLOX-Pro, we compared their landslide detection results. As shown in Figure 12, YOLOX performs poorly in detecting small landslides in complex environments, with more missed detections. YOLOX-Pro has significantly improved the problems above, with few missed detections and strong detection performance.

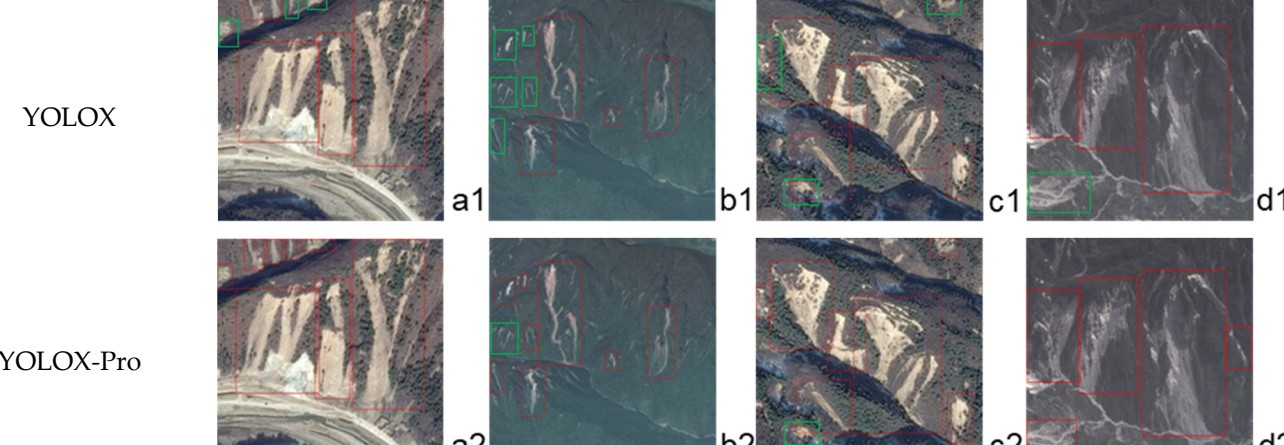

**Figure 12.** Comparison of detection results of YOLOX(M) and YOLOX-Pro(M) models. The red target boxes indicate the landslides automatically detected by the model, and the green target boxes are the manually labeled missed landslides. (**a1**–**c1**): the detection effect of small landslides is poor, and there are many missed detections; d1: the debris flow in the lower left corner of the image was missed; (**a2**–**d2**): better detection performance, least missed detection.

### 4.3. Landslide Detection Results

We present the results of the landslide detection of YOLOX-Pro on the CN test set. Some representative detection results are shown in Figure 13. The red target boxes mark the boundaries of the landslides and the corresponding confidence scores. A higher confidence score means a higher probability that the detected object is a landslide.

The analysis shows that the model can accurately delineate the landslide boundary in the correctly detected area, especially distinguishing between landslides and roads or rivers, as shown in Figure 13a–e. When detecting multiple landslide samples, the model can accurately distinguish landslides of different sizes, as shown in Figure 13f–j. The model still has a high discrimination ability for complex landslide areas when the historical landslide areas are very similar to the surrounding environment or some areas have been covered by surface vegetation, as shown in Figure 13k–o. The detection performance of the model for minor landslides is demonstrated in Figure 13p–t. The model can correctly identify small landslides along the road, which is essential for determining dangerous hidden areas and ensuring transportation safety. In general, the YOLOX-Pro model has high detection accuracy and strong robustness.

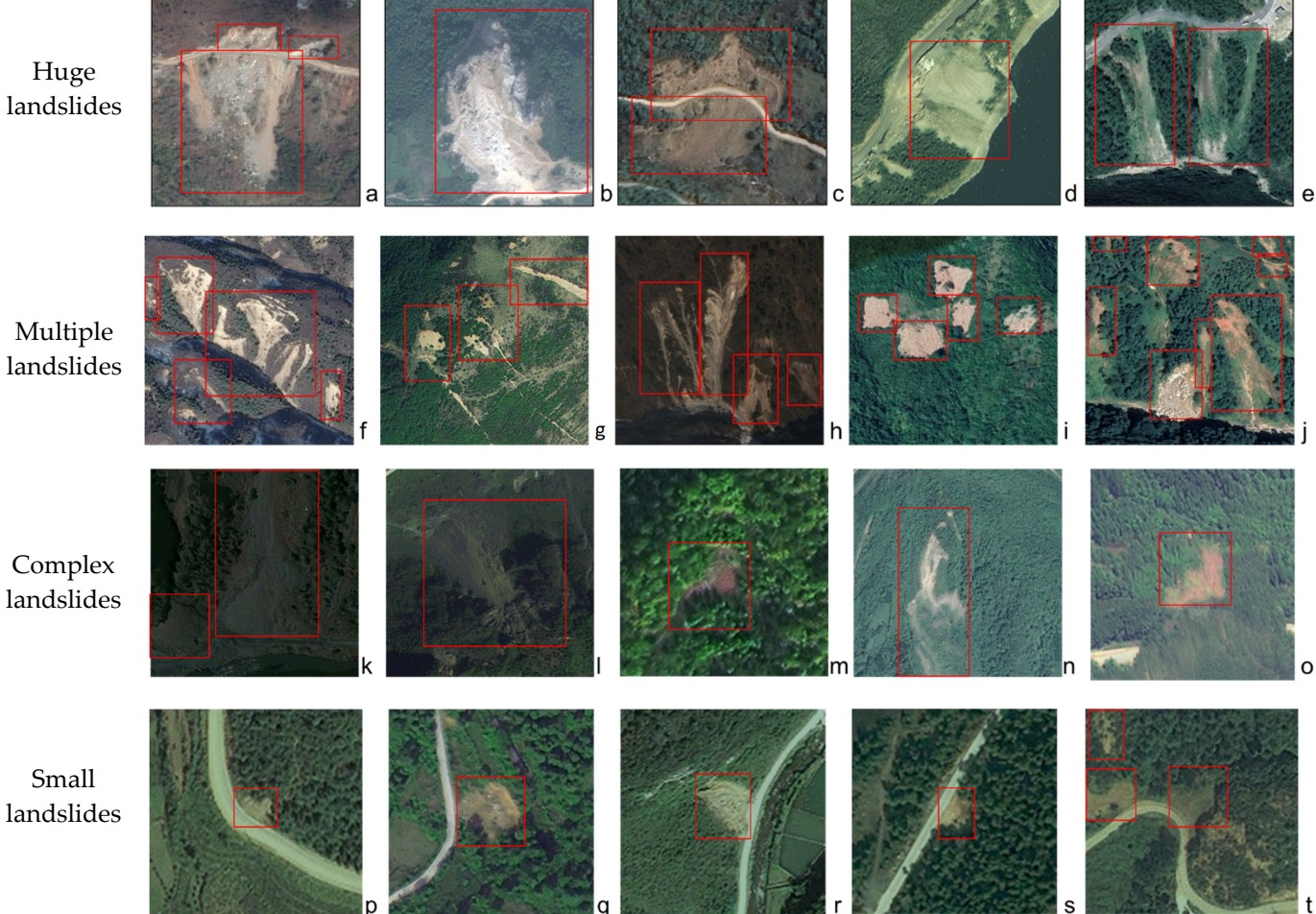

**Figure 13.** Detection results of potential landslide areas by YOLOX-Pro(M). The partially representative detection results of YOLOX-Pro for potential landslide areas. The red rectangles mark the boundaries of the landslide, which the model automatically detects. (Huge landslides) (**a**–**e**): the huge landslides. (Multiple landslides) (**f**–**j**): the multiple landslides of different sizes; (Complex landslides) (**k**,**l**): the historical landslides areas are very similar to the surrounding environment; (**m**–**o**): the landslides that have been covered by surface vegetation in some areas. (Small landslides) (**p**–**t**): the small landslides along the road.

### 4.4. Landslide Detection Results in The Mibei Area

To further measure the detection performance of the proposed model, YOLOX-Pro was used to detect group-occurring landslides in the Mibei area. The detection results of the densest area of the landslide are shown in Figure 14.

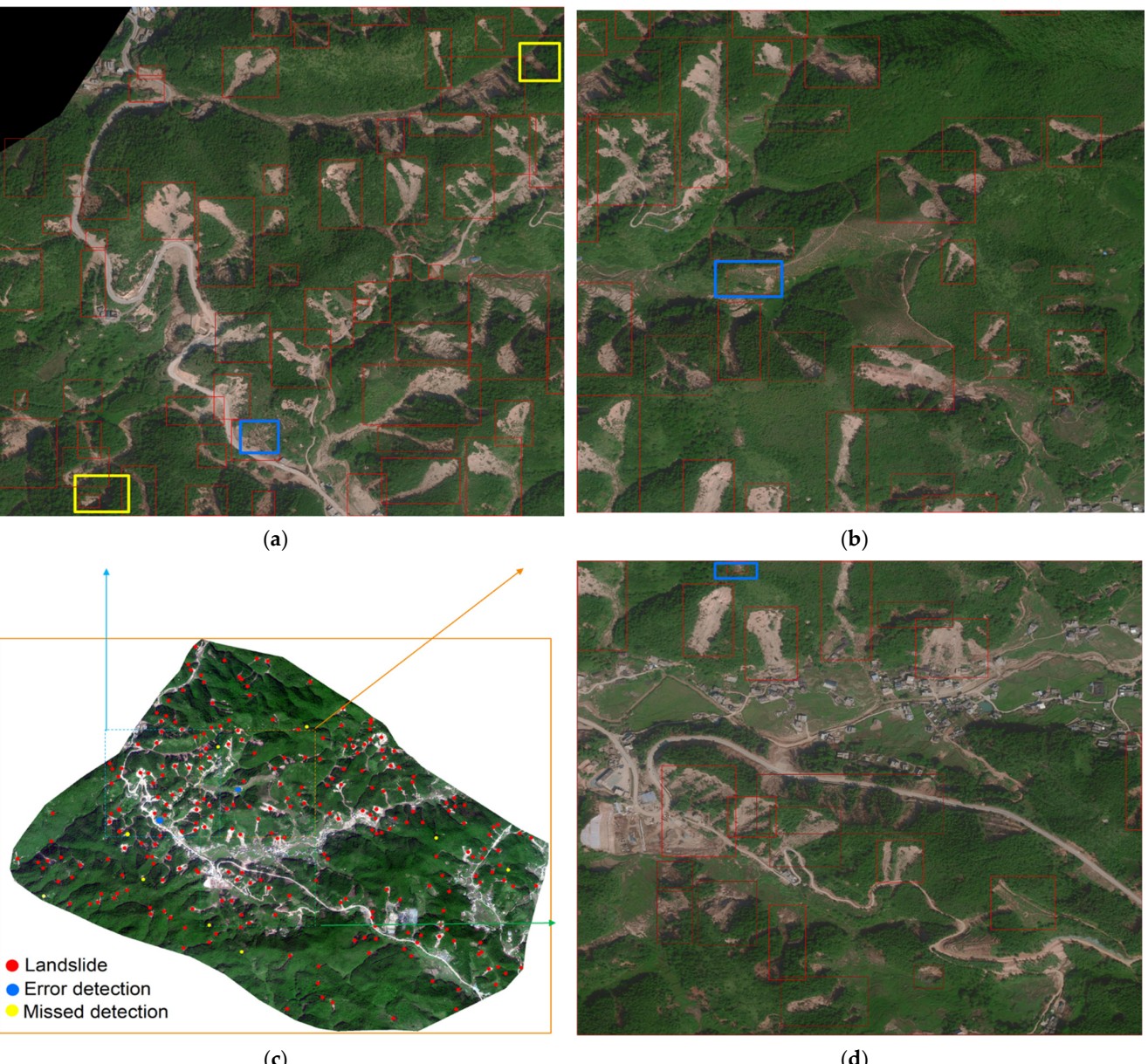

**Figure 14.** Detection results of Mibei area by YOLOX-Pro(M). (**a**,**b**,**d**): Detection results of dense areas of landslides. (**c**): Landslide detection results for the Mibe area. The red, blue, and yellow points indicate the correct, erroneous, and missed landslide detection locations, respectively. The red rectangles mark the landslide boundaries, which the model automatically detects. The yellow and blue rectangular boxes are the model's missed and erroneously detected landslides, respectively.

Figure 14 shows that YOLOX-Pro still has a good detection performance for cluster-type landslides, with only a few missed and false detection results. Part of the landslide in the yellow box in Figure 14a is blocked by the shadows of vegetation, which leads to missed detection. The bare ground eroded by rain in the blue boxes in Figure 14a,b is very similar to the landslide and is incorrectly detected as a landslide. The above two types of samples are added to the training to improve the model's discriminative ability, thus solving the above problem. The bare land on the edge of Figure 14d is incorrectly detected as a landslide. The above problem can be avoided by keeping a certain image cropping threshold, as in Figure 14b, where this bare land is not identified as a landslide.

The results of the YOLOX-Pro network for detecting landslides in the Mibei area are shown in Table 7.

**Table 7.** Results of the YOLOX-Pro network for detecting the Mibei area.

| Model | Params (Mb) | AP0.5 (%) | Recall (%) | Precision (%) |
|---|---|---|---|---|
| YOLOX-Pro (l) | 57.51 | 85.19 | 86.76 | 83.95 |
| YOLOX-Pro (m) | 26.50 | 83.77 | 83.88 | 81.37 |
| YOLOX-Pro (s) | 10.30 | 81.86 | 82.51 | 80.86 |
| YOLOX-Pro (tiny) | 6.51 | 80.55 | 80.22 | 80.35 |
| YOLOX-Pro (nano) | 1.42 | 79.30 | 79.18 | 79.56 |

*4.5. UAV Landslide Detection*

In geological disasters, using UAVs to conduct timely surveys of the affected area is very important for disaster risk evaluation and rescue work. Therefore, we add UAV images for landslide detection.

From Table 8, the results show that all five sizes of YOLOX-Pro networks have high detection performance for UAV landslide images, and the differences are slight, especially the YOLOX-Pro (nano) AP0.5 achieved 82.47%, and the parameters are only 1.42 Mb.

**Table 8.** Results of the YOLOX-Pro network for detecting the UAV dataset.

| Model | Params (Mb) | AP0.5 (%) | Recall (%) | Precision (%) |
|---|---|---|---|---|
| YOLOX-Pro (l) | 57.51 | 86.35 | 83.51 | 84.65 |
| YOLOX-Pro (m) | 26.50 | 85.87 | 82.37 | 83.28 |
| YOLOX-Pro (s) | 10.30 | 84.56 | 81.77 | 82.54 |
| YOLOX-Pro (tiny) | 6.51 | 83.28 | 80.82 | 81.66 |
| YOLOX-Pro (nano) | 1.42 | 82.47 | 80.36 | 80.88 |

The detection results are shown in Figure 15. YOLOX-Pro has good detection performance for many types of landslides and is highly portable.

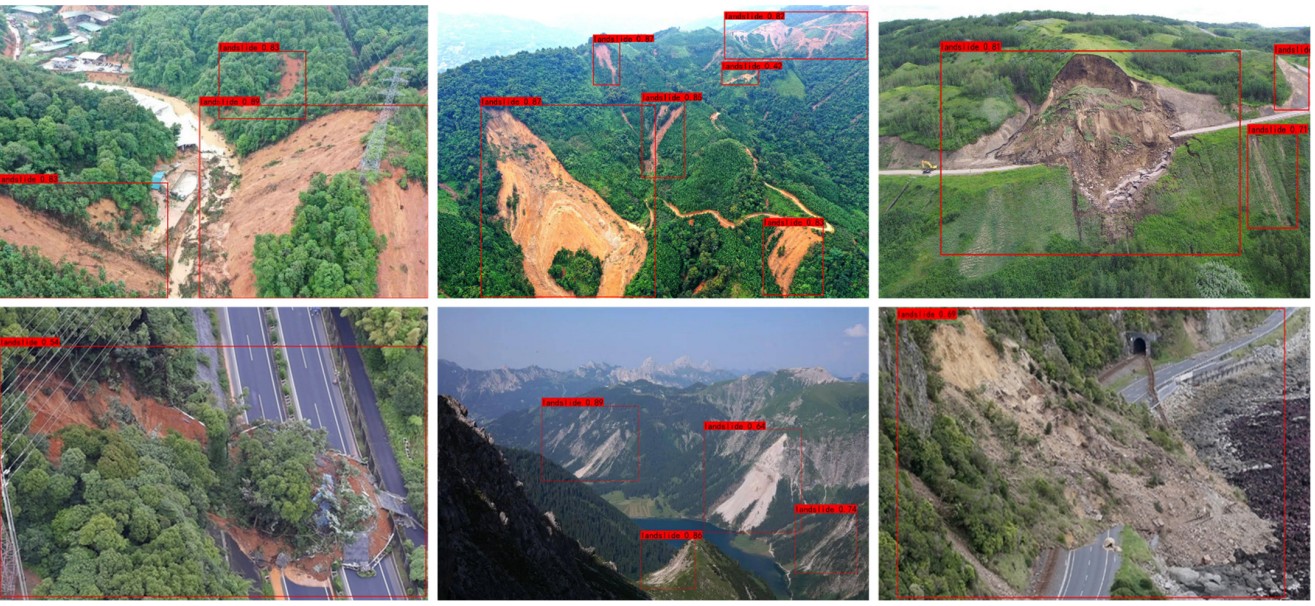

**Figure 15.** Detection results of UAV landslide image area by YOLOX-Pro(M).

This experiment used images taken by UAVs to perform landslide detection on a server, providing a technical reference for using UAVs for large-area landslide detection. With the development of related technologies, deploying lightweight landslide detection

models directly to UAVs for real-time landslide detection is a more rapid method, which is the focus of future research.

## 5. Discussion

Deep learning tasks usually use many data preprocessing operations and tricks to improve model performance [48,59–64]. Nevertheless, it is critical to tailor the appropriate technology to the task requirements. Since this study aims to establish a universal landslide detection method and therefore does not use many tricks to avoid too much intervention.

In this section, the effect of the quantity and location of the CA modules on the base model was first tested. Then, we discuss the limitations and future challenges of this study.

### 5.1. Comparing Different Locations and The Number of Attention Modules

The location and number of attention modules can affect model performance. In this experiment, we add the CA module to the last layer of the CSP block. Three different network locations were tested using different numbers of attention modules, corresponding to the red dashed boxes at the dark2, dark3–5, and Neck locations in Figure 6. The dark2 position uses 1 CA module, and the remaining two positions use 3 CA modules each. Experiments were performed on the M and Nano sizes of the base model, and the results are shown in Table 9.

**Table 9.** The effect of the location and quantity of CA modules on the base model. The YOLOX model with the VariFocal loss function was used as the base model for this experiment.

| Description | Params (Mb) | AP | AP0.75 | APsmall | ARsmall |
|---|---|---|---|---|---|
| base model(m) | 26.50 | 47.50 | 49.50 | 34.90 | 47.80 |
| YOLOX -CA _dark2(m) | 26.50 | 47.80 | 51.50 | 35.80 | 49.50 |
| YOLO -CA _dark3–5(m) | 26.58 | 46.80 | 50.40 | 33.50 | 49.20 |
| YOLOX -CA _neck(m) | 26.55 | 47.40 | 49.90 | 34.70 | 48.50 |
| base model (nano) | 1.40 | 45.50 | 46.60 | 32.50 | 46.70 |
| YOLOX -CA _dark2(nano) | 1.42 | 45.90 | 46.90 | 33.70 | 47.50 |
| YOLOX -CA _dark3–5(nano) | 1.44 | 44.60 | 46.80 | 31.20 | 47.50 |
| YOLOX -CA _neck(nano) | 1.44 | 45.30 | 46.60 | 31.90 | 47.40 |

In the following, the experimental results will be analyzed in two aspects.

Effect of CA module location: Placing the CA module in the dark2 position improves the model's overall performance, with AP, AP0.75, APsmall, and ARsmall improved, whereas at dark3–5 positions decreased AP and APsmall, moreover, placed in the Neck position slightly affects the model performance. The attention module added to the Backbone has a greater effect on the model performance than that added to the Neck because the feature extraction of input mappings is performed in the Backbone, whereas in the Neck, only fusion of the features.

Effect of the number of CA modules: The detection of complex landslides and small landslides would increase their weights since the base model uses the VariFocal loss function. Again, too many attention mechanisms will cause the model to focus too much on a certain area. The overall detection accuracy of the landslide area will be reduced, and overfitting problems will arise. Thus, three CA modules were placed in the dark3–5 and Neck positions, respectively, leading to the model's decrease of AP and APsmall. While only one CA module was placed in the dark2 position, combining the impact of VariFocal, the model's performance was improved.

### 5.2. Limitations and Future Challenges

When using images for landslide detection, some landslides were difficult to observe directly due to the influence of the shooting angle, resulting in missed detection. For

example, the landslides in the blue box in Figure 16b and the orange box in Figure 16c were missed. However, the landslide features are more obvious after switching the shooting angle, and the model can detect them (the landslide in the blue box in Figure 16a). Therefore, using multi-angle images for landslide detection can improve detection accuracy.

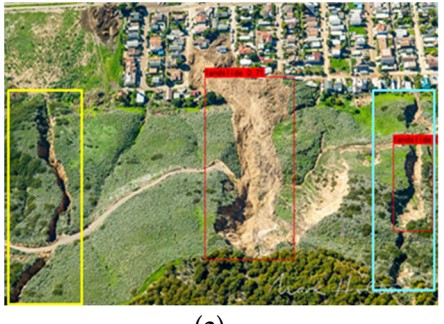 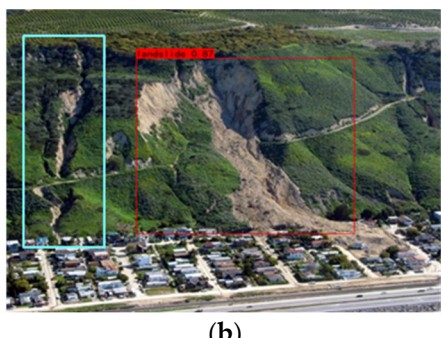 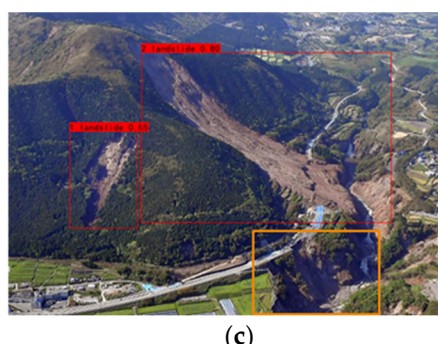

(**a**) (**b**) (**c**)

**Figure 16.** (**a**): The yellow box is the landslide fissure, and the blue box is the landslide detected after converting the angle; (**b**): the blue box is the missed landslide; (**c**): the orange box is the missed landslide.

The UAV can capture high-definition images of the ground at close range. Therefore, subsequent research can use image recognition technology to detect fissures automatically generated by landslides, which is essential for pre-disaster warning of landslides and determination of landslides (e.g., the landslide fissures in the yellow box in Figure 16a).

To achieve automatic and fast detection of landslides, the YOLOX-Pro model uses rectangular boxes to label the landslide area but fails to obtain more detailed information about the landslide. Subsequent research focuses on using image segmentation algorithms to obtain irregular bounding boxes of landslide areas, calculate their areas and obtain landslide coordinates. In addition, more categories of landslides can be collected for multi-category labeling to achieve automatic multi-category landslide detection. The above studies are crucial to obtain more landslide hazard information and quickly assess landslide hazard risk.

The use of deep learning techniques for landslide hazard detection is still in the preliminary stage of application. Nevertheless, the method demonstrates good detection performance when sufficient landslide data are obtained and has high application value and broad development prospects. Future research can explore the combination of landslide area-related data (e.g., Digital Elevation Model (DEM)) to improve landslide detection accuracy. Based on the rapidly developing artificial intelligence technology, it is of high research significance and value to combine the massive landslide disaster-related information (such as extreme weather warning, earthquake warning, etc.) for pre-disaster warning of landslides. Ultimately, establishing a complete and efficient pre-disaster warning and post-disaster analysis system for landslides is both a challenge and an opportunity for researchers in related fields.

## 6. Conclusions

This paper established a dataset containing multiple types of remote sensing landslides using Google Earth images as the data source. Based on the deep learning model YOLOX, a set of YOLOX-Pro networks containing five scales is constructed by replacing the VariFocal loss function and introducing a lightweight Coordinate Attention mechanism to propose a universal detection method for multi-category landslides. The model is highly portable and can be used by selecting a network of the corresponding size according to the detection equipment's storage and computational capacity status, which has broad application prospects. Then, to evaluate the model's capability in detail, the COCO evaluation metrics were the first introduced into the landslide detection task. The results show that the proposed method has high detection accuracy for small and complex mixed

landslides. Finally, the detection results of the group-occurring landslide area and the UAV landslide images verify the robust performance of the proposed model. In addition, we discuss the limitations of this study and challenges for future research, which provides a technical reference for further exploration of landslide hazard research using deep learning techniques. Since the dataset is small, the proposed method still has some error detection, which is an issue to be addressed in later work. In future work, we will investigate using semantic segmentation algorithms to obtain more information about landslides for hazard surveys and explore deploying lightweight models to embedded devices to detect landslides automatically.

**Author Contributions:** H.H. drafted the manuscript and was responsible for the research design, collected the image data, experimented, and analyzed. M.C. provided funding support and reviewed and edited the manuscript. Y.T. provided ideas and reviewed and edited the manuscript. W.L. provided technical guidance. All of the authors contributed to editing and reviewing the manuscript. All authors have read and agreed to the published version of the manuscript.

**Funding:** This research was supported by the National Natural Science Foundation of China (No. 61863009).

**Institutional Review Board Statement:** Not applicable.

**Informed Consent Statement:** Not applicable.

**Data Availability Statement:** The datasets generated and/or analyzed during the current study are available from the corresponding author on reasonable request.

**Conflicts of Interest:** The authors declare no conflict of interest.

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
