# Peer review of "A Universal Landslide Detection Method in Optical Remote Sensing Images Based on Improved YOLOX"

_remotesensing, doi:10.3390/rs14194939_

Round 1

Reviewer 1 Report (Previous Reviewer 1)

Although I did not find the detail resposne to reviewer, but I could find the revision from the revised manuscript.The authors addressed most of my comments. I am satistifed with the revision. I think it could be accepted for publicaiton.

Author Response

Response to Reviewer 1 Comments

Comment: Although I did not find the detail resposne to reviewer, but I could find the revision from the revised manuscript. The authors addressed most of my comments. I am satistifed with the revision. I think it could be accepted for publicaiton.

Response:

Dear Reviewers,

Thank you very much for reviewing our manuscript and providing us with valuable suggestions for improvement. These comments were very valuable and very helpful in improving our paper.

In the last revised manuscript, we addressed the first comments and made significant revisions. We are very sorry that you did not receive a detailed response letter to your comments.

Finally, we sincerely thank you for your recognition of our manuscript, which made it possible to be published.

Reviewer 2 Report (New Reviewer)

Reviewer’s Report on the manuscript entitled:

A Universal Landslide Detection Method Research in Optical Remote Sensing Images Based on YOLOX-Pro

The authors developed landslide detection models based on the improved YOLOX target detection model to address the poor detection of complex mixed landslides. I found the topic and results interesting. I have however a few comments listed below.

Please check that all the abbreviations are defined the first time they appear.

Lines 28-29. Grammar issue. “…with an average precision (AP0.75) is…”. Please rewrite.

Line 32. Please replace “12.32-square-kilometers” with “12.32 km2”

Line 68. Please remove the hyphen for “time-series”. It is recommended to be “time series”. Also, please include the following reference describing jump/change detection algorithms and software within remote sensing satellite data:

https://doi.org/10.1007/s10291-021-01118-x

and also the following article for landslide detection through Sentinel-2 time series:

https://doi.org/10.1007/s10346-019-01178-8

Line 105. Please say instead “The main contributions of this work are as follows.”

Line 124. Please say “…points distributed as shown in Figure 1.

Figure 3. It is not clear which of the images refer to Landslide samples, which ones refer to CN negative samples, etc. Please use (a), (b) and (c) for the set of each 10 panels and write clearly in the caption.

Line 195. The sigmoid function is mathematically incorrect. Please use the parentheses properly.

Equation (5), (6), and (7). Please describe all the parameters.  

Line 702. This is also true when using the InSAR satellite data. InSAR imagery with ascending and descending geometries can be used together for landslide detection. You may also add this article here:

https://doi.org/10.1038/s41598-022-06508-w

Please follow the MDPI guidelines for formatting all the references. Pleased check the authors names, year of publication, volume, page number or article number carefully.

Please carefully proofread the manuscript.

Thank you for your contribution

Regards,

Author Response

Reviewer 3 Report (New Reviewer)

I feel Figure 6 is not well-described. The reader can not follow.

For instance, what is CBS?

The paper needs a parallel comparison of with/without COCO.

Why the paper used YOLO? Can the problem be solved by GAN or CNN? it need to be explained.

Author Response

This manuscript is a resubmission of an earlier submission. The following is a list of the peer review reports and author responses from that submission.

Round 1

Reviewer 1 Report

Hou et al. constructs a set of landslide detection models YOLOX-PRO based on the YOLOX serial models. The YOLOX-PRO has improved detection accuracy over previous models. To be honest, I am not familiar with the models the authors mentioned in the manuscript. But it seems that the working flow and the model parameter are sound and reasonable. It should be eventually published.

The main comments are as following:

1.     It seems the authors using 1200 selected landslides to test the accuracy of the YOLOX-PRO model. Why choose these 1200 landslides? The criteria for selection should be introduced.

2.     The authors use the images covering landslide region to test YOLOX-PRO model. This means the target images are already provided. If without such input images of the landslides, would it be possible for YOLOX-PRO to distinguish landslides from non-landslide region like change detection methods using pre-landslide and post-landslide images?

3.     Why the training imagery dataset is limited to 640*640 resolution? It seems too small for broader application of the YOLOX-PRO model.

4.     It seems the detection results just show there are landslides as shown in the red rectangles in the figures. Will YOLOX-PRO provide vector polygon results of the landslides regions?

Reviewer 2 Report

The authors propose a large improvement of the automatic methods to detect landslides by diminishing the number the preprocessing phases, and simplify the use. They test and improve a YOLOX method based on neuronal algorithms and compare the results with dataset of landslides coming from interpretation of Google Earth images and from drone images.

The manuscript is reserved of high-specialist of neuronal methods and difficult to follow in some parts. For instance, the authors cite a lot of methods in the introduction by never shortly described them. Thus, in this form, I propose to resubmit it in other journal.

In terms of results, the figure 12 perfectly sums up the “efficiency” of the method that missed the landslides that have not “regular” shapes, i.e. linear and roughly parallel borders. Maybe, it comes from a step in coordinate Attention workflow during which the cubic data are split into into 2D matrix.

The tables showing the different comparisons (tables 3, 4, 5 , 6, 7, 8) are hard to understand because the units of the values are not given. It is probably percentages, but the reference to estimate them is not given. Thus, it is impossible to convince the reader.

This manuscript suffers also the lack of maps showing the landslides detected. Only snapshots are presented with red rectangles. Does the processing provides these rectangles or is there more pertinent information? It is really impossible to realize concretely the efficiency of the method and how to use the results.

At least, the authors often announce some facts without any argues. This sounds a lot like more allegations with no real basis, see examples in the section of detailed comment and in the annotated file.

To conclude, I am not convinced about the efficiency of the proposed method, although it is a lot of work. I suggest to reject this manuscript and I propose to submit the article in a journal more specialised into numerical methods.

Olivier Dauteuil

Detailed comments:

L42: explain why it is inefficient.

L47: explain longer this assertion.

L57: mean ing and explanation of CNN-based detection algorithms 

In the text, there are a lot of acronyms that are not explain.

L93: the first step of the method needs a dataset of landslides that was done from Google Earth. How this dataset was built? By eyes recognition? Explain clearly because this is an important step. Were the negative samples removed? Why there are negative sample? 

The figure 2 shows some examples of landslides seen on Google Earth and on UAV. The 3rd and 4th raws indicate negative samples. Why are they negative? Please explain.

L111: add  reference ce and explain in 2-3 sentences the principle of this method.

L186: list them now to better introduce the following sections

the section describing the YOLOX-Pro algorithm is not enough commented. It remains too simple, while it should be one of the most important improvement of this work. The figure 7 that should illustrate the improvements of data augmentation explain nothing! And the text never described the improvement.  The authors should longer explain the benefits to be credible.

I don’t see where datasets are introduced during the processing.

Section 3.4.1 - CA

L260: I don’t understand the aggregation one-dimension features. Please explain longer.

L294: explain the coco evaluation method.

L297: flexibility. What do you mean.

L370: units? do you mean 32 pixels x 32 pixels ? You should indicate an area with length unit: meters, kilometers ….

The table 3 is important because it compares the performance of the different method. However, the meaning of the values in the table is not indicating. So it is impossible to understand it. So the conclusion given in the next section is impossible to verify! 

Figure 10: this figure displays some examples of the detected landslides. What do the red rectangles mean? Is it the result of the complete processing? I.e. does the processing provide rectangles where landslides are detected? If yes, how the landslide area is estimated? The lower row displays errors in detection, it is a very good idea to add this. How do you know that it is an error? How did you check them? From dataset? I am not convinced that the example 10u 10y are an error. 

The authors should indicate the unit of the values in the tables 3,4,5,6 and 7. I think that it is percentage. If yes, you should indicate the reference used to estimate the percent.

L440-450: what is the field evidences of landslides detected.

Reviewer 3 Report

The purpose of this article is to present the capability of Optical Remote Sensing Images Based on YOLOX-Pro in landslide detection. The article is well structured and clear in its purpose; the results are interesting and promising. The paper can be published in Remote  Sensing. I have just few recommendations:

- improve a little the discussion related to the comparison with other methods/approaches available in literature

- add a sub-chapter entitled "limitations & future challenges" with half a page text, this will help the readers to better understand the potential of the work.